# Non-Inferiority Field Study Comparing the Administrations by Conventional Needle-Syringe and Needle-Free Injectors of a Trivalent Vaccine Containing Porcine Circovirus Types 2a/2b and *Mycoplasma hyopneumoniae*

**DOI:** 10.3390/vaccines10030358

**Published:** 2022-02-24

**Authors:** Hyejean Cho, Yongjun Ahn, Taehwan Oh, Jeongmin Suh, Chanhee Chae

**Affiliations:** Department of Veterinary Pathology, College of Veterinary Medicine, Seoul National University, Gwanak-ro 1, Gwanak-gu, Seoul 08826, Korea; hcho21@snu.ac.kr (H.C.); ayj3ca@gmail.com (Y.A.); ohth93@gmail.com (T.O.); tobin1210@snu.ac.kr (J.S.)

**Keywords:** *Mycoplasma hyopneumoniae*, needle-syringe, needle-free injector, porcine circovirus type 2, vaccine

## Abstract

The objective of this study was to assess the clinical, immunological, microbiological, and pathological evaluation of trivalent vaccine containing porcine circovirus types 2a/b (PCV2a/b) and *Mycoplasma hyopneumoniae* given by two different needle-free injection devices compared with conventional needle-syringe injection in a herd with subclinical PCV2d infection and enzootic pneumonia. A total of 240 21-day-old pigs, which weighed between 5 to 6 kg, were randomly divided into four groups (60 pigs per group, 30 = male and 30 = female per group). Injection site reactions in the pigs were minimal for the two needle-free injection devices and needle-syringe injection. Trivalent vaccination of pigs with two needle-free injection devices was not inferior to conventional needle-syringe injection for growth performance. Trivalent vaccination of pigs with two different needle-free injection devices reduced levels of PCV2d loads in serum and *M. hyopneumoniae* loads in the larynx equally compared to the conventional needle-syringe injection. The amount of PCV2d load in serum from the needle-free Pulse FX injection device at 49 days post vaccination showed non-inferiority to conventional needle-syringe injection. The immune response against PCV2 and *M. hyopneumoniae* to trivalent vaccine given with the needle-free Pulse FX injection device was non-inferior to conventional needle-syringe injection. The pigs from the two needle-free injection device and conventional needle-syringe injection had significantly (*p* < 0.05) lower macroscopic and microscopic lung lesion scores, and microscopic lymphoid lesions than from unvaccinated. The results of this study demonstrated that vaccination of trivalent vaccine by the two needle-free Pulse injection devices used in the study was non-inferior to that by conventional needle-syringe injection for growth performance, immune response against PCV2 and *M. hyopneumoniae*, and reduction of PCV2 viremia.

## 1. Introduction

Porcine circovirus type 2 (PCV2) and *Mycoplasma hyopneumoniae* are two major economically important pathogens in global pork industry. PCV2 is the principal causative agent of porcine circovirus-associated disease (PCVAD), which was originally described as postweaning multisystemic wasting syndrome [1,2]. Subclinical PCV2 infection is currently the most common form of PCVAD [3]. Additional disease manifestations such as porcine respiratory disease complex (PRDC), and reproductive failure have been found in all ages of swine [1]. *M. hyopneumoniae* lacks a cell wall and is one of the smallest bacteria found in nature. As a primary contributor to PRDC, *M. hyopneumoniae* is the etiological agent of enzootic pneumonia [4]. Although it is not a new pathogen, for these reasons, it is one of the most economically devastating pathogens to the swine industry.

PCV2 and *M. hyopneumoniae* play an important role in PRDC [4,5]. Control of these two pathogens is heavily dependent on vaccination. Immunization via needle-syringe injection has been in practice since the origin of vaccinology in veterinary medicine, but such conventional vaccination presents few challenges, as it is time-consuming for swine workers and stressful for pigs, as fear and pain indicators at the time of injection are increased [6]. Needle-free injection devices offer advantages over conventional needle-syringe injection, such as increased safety of swine workers, less pain and stress of pigs, avoidance of broken needles, and less risk of iatrogenic pathogen transmission between pigs caused by reusing needles [6,7,8,9,10]. The elimination of broken needles and associated trim at the process is the most important of these elements in pork quality assurance. Consequently, interest in the use of a needle-free device system has increased over the last three years in the Asian and global pig industry.

A new trivalent PCV2a/b and *M. hyopneumoniae* vaccine (Fostera^®^ Gold PCV MH /CircoMax^®^ Myco, Zoetis, Parsippany, NJ, USA) has recently been introduced into the market [11]. The PCV2b antigen in this trivalent vaccine is genetically similar to PCV2d, which was formerly referred to as ‘mutant PCV2b’. Currently, PCV2d is the most prevalent field genotype spread worldwide [12,13,14]. Despite the advantages of needle-free injection devices over conventional needle-syringe injection, clinical trials on needle-free injection devices have not yet been reported for this new vaccine. The objective of this study was to compare the effectiveness of two different needle-free injection devices with conventional needle-syringe used to administer a trivalent PCV2a/b and *M. hyopneumoniae* vaccine in a pig herd suffered from subclinical PCV2d infection and enzootic pneumonia, in relation to growth performance, induction of PCV2 and *M. hyopneumoniae* antibodies, reduction of PCV2d viremia and mycoplasmal laryngeal shedding, and the reduction of pathological lesions under field conditions.

## 2. Materials and Methods

### 2.1. Farm

The clinical field trial was conducted on a 1200-Large White × Landrace cross bred sow farrow-to-finish swine farm with an all-in-all-out production system. Farm selection was based on clinical history of subclinical PCV2 infection and enzootic pneumonia. The chosen farm status of porcine reproductive and respiratory syndrome (PRRS) was seropositive but absence of clinical signs of PRRS in breeding-herd population. A detailed clinical history of the farm was described elsewhere [15].

### 2.2. Experimental Design

A total of 240 21-day-old pigs, weighing 5 to 6 kg each, were randomly divided into four groups (60 pigs per group, 30 = male and 30 = female per group) using the random number generator function (Excel, Microsoft Corporation, Redmond, Washington, DC, USA). The pigs in the VacS, VacPulse, and VacEPIG groups were intramuscularly vaccinated with a 2.0 mL dose of Fostera Gold PCV MH (Serial no: 413369A, Expiration date: 03-Feb-2022, Zoetis, Parsippany, NJ, USA) by either conventional needle-syringe injection (VacS group), needle-free Pulse FX injection device (Serial no. FX-200-0111, Pulse NeedleFree System Inc., Lenexa, KS, USA) (VacPulse group), or by needle-free EPIG injection device (Serial no. 1553569, Henke-Sass Wolf GmbH, Tuttlingen, Germany) (VacEPIG group), respectively, at 21 days of age. Conventional needle-syringe-based vaccination was administered intramuscularly with a pistol-grip syringe fitted with a 10 mm, 22-gauge needle (KOVAX-SYRINGE, Korean Vaccine, Seoul, Republic of Korea). Compressed CO_2_ was used as the power source for the needle-free Pulse FX injection device. Using guidelines from Pulse Needle Free Systems Inc, the device was set at a pressure of 85 to 95 pounders per square inch (PSI) on 21 days old pigs to deliver the intramuscular injection of 2.0 mL per pig. The needle-free EPIG injection device was set at the default setting of a 2.0 mL dosage. The pigs in the UnVac group were intramuscularly vaccinated with a 2.0 mL dose of phosphate buffered saline (PBS, 0.01 M, pH 7.4) by conventional needle syringe at 21 days of age (Table 1).

Pigs were comingled with treatment groups and randomly assigned into 12 pens within the same building. Each pen contained 20 pigs with a same proportion of each treatment per pen (five pigs per group). Pens were identical in design and equipment, which included free access to a feed and water trough. Blood and laryngeal swabs were collected at 0 (21 days of age), 28 (49 days of age), 49 (70 days of age), and 91 (112 days of age) days post-vaccination (dpv). All the methods were approved by the Seoul National University Institutional Animal Care and Use, and Ethics Committee (SNU-210518-3).

### 2.3. Post-Vaccination Skin Reaction

Presence of vaccine residue at the skin surface was recorded immediately following vaccination. Pigs were visually scored for the presence of post-vaccination skin reactions at 0, 1, 4, 7, 14, and 21 dpv. Any raised surface observed at the injection site was considered a skin reaction occurring as a result of vaccination.

### 2.4. Clinical Observations

The pigs were monitored daily for abnormal clinical signs and scored weekly using scores ranging from 0 (normal) to 6 (severe dyspnea and abdominal breathing) [16]. Observers were blinded to vaccination and type of vaccine status. Mortality rate was calculated as the number of pigs that died divided by the number of pigs initially assigned to that group. Pigs that died or were culled throughout the study were necropsied.

### 2.5. Average Daily Weight Gain

The live weight of each pig was measured at 0 (21 days of age), 49 (70 days of age), and 154 (175 days of age, or slaughter day) days post-vaccination. The average daily weight gain (ADWG; grams/pig/day) was analyzed over two time periods: (i) between 21 and 70, and (ii) between 70 and 175. ADWG during the different production stages was calculated as the difference between the starting and final weight divided by the duration of the stage. Data of body weight for dead or removed pigs were included in the calculation.

### 2.6. Serology

The serum samples were tested using the commercially available PCV2 (INgezim CIRCO IgG, Ingenasa) and *M. hyopneumoniae* (*M. hyo.* Ab test, IDEXX Laboratories Inc., Westbrook, ME, USA) ELISA kits. Samples were considered positive for PCV2 antibodies if the optical density (OD) was >0.3 and for *M. hyopneumoniae* antibodies if the sample-to-positive (S/P) ratio was ≥0.4, according to the manufacturer’s instructions.

### 2.7. Quantification of PCV2d DNA in Serum

DNA was extracted from serum samples using the commercial kit (QIAamp DNA Mini Kit, QIAGEN, Valencia, CA, USA) to quantify PCV2d genomic DNA copy numbers by real-time PCR [2].

### 2.8. Quantification of M. hyopneumoniae DNA in Larynx

DNA was extracted from laryngeal swabs using the commercial kit (QIAamp DNA Mini Kit, QIAGEN) to quantify *M. hyopneumoniae* DNA copy numbers by real-time PCR [17].

### 2.9. Pathology

The severity of macroscopic lung lesions was scored to estimate the percentage of the lung affected by pneumonia. The scoring was done by two pathologists (Chae and one graduate student) at the Seoul National University (Seoul, Republic of Korea). For the entire lung, 100 points were assigned as follows; 10 points each to the right cranial lobe, right middle lobe, left cranial lobe, and left middle lobe, 27.5 points each to the right caudal lobe and left caudal lobe, and 5 points to the accessory lobe [16]. Two blinded veterinary pathologists then examined the collected lung and lymphoid tissue sections and scored the severity of peribronchiolar lymphoid tissue hyperplasia by mycoplasmal pneumonia lesions (0 to 6) [18]. Lymphoid lesion severity was scored (0 to 5) based on lymphoid depletion and granulomatous inflammation [19].

### 2.10. Statistical Analysis

Prior to statistical analysis, real-time PCR data were transformed to log_10_ values. The Shapiro–Wilk test was used to test the collected data for a normal distribution. Results were reported in *p*-Value, where a value of *p* < 0.05 was considered significant.

ADWG, PCV2 and *M. hyopneumoniae* ELISA, and real-time PCR PCV2 and *M. hyopneumoniae* DNA quantification data were analyzed using a generalized linear mixed model for repeated measures with the fixed effects which includes treatment, time point, and their interaction. As the random effects, pen and animal are included. Each time point estimation has the least squares means (or geometric means for PCR data and ELISA titer) and the relative 95% confidence intervals (CI) limits. For ADWG, PCV2 and *M. hyopneumoniae* ELISA, and real-time PCR PCV2 and *M. hyopneumoniae* DNA quantification results, the non-inferiority trial was conducted to compare between needle-free injection (VacPulse and VacEPIG) groups within the reference group, the conventional needle-syringe injection (VacS) groups. At the time points, 49 (70 days of age) and 91 (112 days of age), the reference group (VacS) was at the 2.5% level for one-sided tests, and the least squares means (delta 10%).

Macroscopic lung lesions (scores between 0 and 100), microscopic lung (scores between 0 and 6) and microscopic lymphoid (scores between 0 and 5) lesions, and mortality were analyzed using a generalized linear mixed model for binomial data with the fixed effect of treatment. The model includes the random effect of pen. The optimal cutoff value for binary outcomes of macroscopic lung lesions, microscopic lung and microscopic lymphoid lesions were determined as >20, >1 and >1, respectively. The back transformed least squares means and their 95%CI were provided. A treatment comparison was made between the needle-free injection (VacPulse and VacEPIG) groups and the reference (VacS) group when the treatment effect is significant in the model.

Clinical signs (normal = 0/abnormal l > 0) were analyzed using a generalized linear mixed model with repeated measurements for binomial data with the fixed effect of treatment, time point, and their interaction. The model includes the random effects of pen and animal. The back transformed least squares means and their 95%CI were provided at each time point. The treatment comparison was made between the needle-free injection (VacPulse and VacEPIG) groups and the reference (VacS) group if any of treatment related effects were significant in the model.

## 3. Results

### 3.1. Post-Vaccination Skin Reactions

Substantial amounts of visible vaccine residue remained on the skin at the site of vaccination for both needle-free injection devices; 30%, or 18/60 pigs from the VacPulse group and 20%, or 12/60 pigs from the VacEPIG group. No visible vaccine residues were not seen in pigs from the VacS group. The injection site reactions were minimal for all vaccine administration types, and injection site reactions that were identified were small swelling, ranging 2–3 in diameter, and self-resolved within 7 days after vaccination. Needle-free devices produced small scars in the injection site area in 4 out of 60 pigs from the VacPulse group and 5 out of 60 pigs from the VacEPIG group.

### 3.2. Clinical Signs

Respiratory sign scores were significantly lower (*p* < 0.05) in the vaccinated pigs (VacS, VacPulse, and VacEPIG groups) than those in unvaccinated pigs (UnVac group) at 28 to 98 dpv. Respiratory sign scores of needle-free injection device-vaccinated pigs in the VacPulse and VacEPIG groups were not statistically different from those of the needle-syringe injection-vaccinated pigs within the VacS group.

### 3.3. Mortality

One 82-day-old pig in the VacS group died of bronchopneumonia as diagnosed by isolation of *Pasteurella multocida* and *Glaesserella parasuis* in the lung. One 77-day-old pig in the VacPulse group died of bronchopneumonia as diagnosed with detection of *M. hyopneumoniae* by PCR and isolation of *G. parasuis* in the lung. Two pigs aged 67 and 72 days in the VacEPIG group died of bronchopneumonia as diagnosed by the detection of *M. hyopneumoniae* with PCR and isolation of *Trueperella pyogenes* in the lung. A total of five pigs died in the UnVac group; three pigs aged 59, 62, and 93 days died of bronchopneumonia as diagnosis by detection of PCV2d and *M. hyopneumoniae* with PCR, and isolation of *P. multocida* in the lung; two pigs aged 71 and 82 days died of bronchopneumonia as diagnosis by detection of *M. hyopneumoniae* with PCR, and isolation of *P. multocida* in the lung.

### 3.4. Average Daily Weight Gain

Body weights were not significantly different at the beginning of study (21 days of age), but were statistical different at the end of the study (175 days of age) between the three vaccinated (VacS, VacPulse, and VacEPIG) and the unvaccinated (UnVac/UnCh) group. The ADWG of vaccinated pigs in the VacS, VacPulse, and VacEPIG groups was significantly higher (*p* < 0.05) than that of unvaccinated pigs in the UnVac group during the fattening period (70 to 175 days of age) and overall period (21 to 175 days). There was no statistical difference in ADWG between vaccinated groups (Table 2).

For the ADWG, during fattening period (70 to 175 days) and overall period (21 to 175 days), the lower bound of the one-sided 95%CI were greater than Delta (the 100% margin of non-inferiority), indicating that the effectiveness of administration via both needle-free injection devices were non-inferior to effectiveness of injection via conventional needle-syringe (Figure 1).

### 3.5. Quantification of PCV2d DNA in Serum

The amount of PCV2d DNA measured in the serum from needle-free device-vaccinated pigs (VacPulse and VacEPIG groups) was significantly (*p* < 0.05) lower at 28 dpv than that of unvaccinated pigs in the UnVac group. The amount of PCV2d DNA in serum of vaccinated pigs from the VacS, VacPulse, and VacEPIG groups was significantly (*p* < 0.05) lower at 49 and 91 dpv than that measured for unvaccinated pigs in the UnVac group (Figure 2).

The amount of PCV2d DNA in serum from the needle-free device (VacPulse) group at 49 dpv showed non-inferiority to conventional needle-syringe injection (VacS group) (95%CI −0.44 to 0.20, non-inferiority margin 0.29). The upper limits of the one-sided 95% CI on the difference in the amount of PCV2d DNA in the serum from the VacPulse and VacEPIG groups at 91 dpv (95%CI for the VacPulse group −0.41 to 0.46 and the VacEPIG group −0.27 to 0.61) exceeded non-inferiority margin (0.24), indicating the outcome did not show non-inferiority (Figure 3).

### 3.6. Quantification of M. hyopneumoniae DNA in Larynx

The *M. hyopneumoniae* DNA load measured in the larynx of needle-syringe-vaccinated pigs in the VacS group was significantly (*p* < 0.05) lower at 28 dpv than that measured for unvaccinated pigs from the UnVac group. The *M. hyopneumonia* DNA loads in the larynx of vaccinated pigs from the VacS, VacPulse, and VacEPIG groups were significantly (*p* < 0.05) lower at 49 and 91 dpv than that of unvaccinated pigs in the UnVac group (Figure 4).

The *M. hyopneuomniae* DNA loads in the larynx of two needle-free injection devices at 49 dpv (95%CI for VacPulse −0.55 to 0.44 and for VacEPIG −0.35 to 0.35, non-inferiority margin 0.18) and 91 dpv (95%CI for VacPulse −0.30 to 0.41 and for VacEPIG −0.25 to 0.46, non-inferiority margin 0.12) did not show non-inferiority to conventional needle-syringe injection.

### 3.7. PCV2 Serology

PCV2 antibody titers of vaccinated pigs in the VacS, VacPulse, and VacEPIG groups were significantly (*p* < 0.05) higher at 28, 49 and 91 dpv than those measured for unvaccinated pigs in the UnVac group (Figure 5). The lower bound of the 95%CI on the difference in PCV2 ELISA at 49 dpv (95%CI for VacPulse −0.07 to 0.03 and for VacEPIG −0.09 to 0.01) did not exceed non-inferiority margin (−0.10). This represents the outcome that shown non-inferiority, but not superiority. The PCV2 antibody titers in the VacPulse group at 91 dpv showed non-inferiority to conventional needle-syringe injection (VacS) group (95%CI −0.10 to 0.04, non-inferiority margin −0.15) (Figure 6).

### 3.8. Mycoplasma hyopneumoniae Serology

*M. hyopneumoniae* antibody titers of vaccinated pigs in the VacS, VacPulse and VacEPIG groups were significantly (*p* < 0.05) higher at 28, 49 and 91 dpv than those measured for unvaccinated pigs in the UnVac group (Figure 7). The lower bound of the 95%CI on the difference in *M. hyopneumoniae* antibody titers at 49 dpv (95%CI for VacPulse −0.05 to 0.10 and for VacEPIG −0.04 to 0.11) did not exceed non-inferiority margin (−0.06). This represents the outcome that shown non-inferiority, but not superiority. The *M. hyopneumoniae* antibody titers in the VacPulse group at 91 dpv showed non-inferiority against the VacS group (95%CI −0.07 to 0.06, non-inferiority margin −0.08) (Figure 8).

### 3.9. Pathology

The pigs from the vaccinated group (VacPulse, VacEPIG and VacS) had significantly (*p* < 0.05) lower macroscopic and microscopic lung lesion scores than from pigs in the UnVac group (Table 3). Compared to the VacS group, macroscopic lung lesions had an OR = 0.96 (95%CI 0.47 to 2.00) for the VacPulse group and an OR = 1.19 (95%CI 0.57 to 2.46) for the VacEPIG group. The VacPulse and VacEPIG groups had higher odds of microscopic lung lesions (VacPulse OR = 11.34, 95%CI 0.63 to 2.89; VacEPIG OR = 1.50, 95%CI 0.70 to 3.24) compared to the VacS group, but they were not statistically different (Table 4).

The pigs in the vaccinated groups (VacPulse, VacEPIG, and VacS) had significantly (*p* < 0.05) lower microscopic lymphoid lesion scores than pigs in the UnVac group (Table 3). Compared to the VacS group, microscopic lymphoid lesions had an OR = 0.56 (95%CI 0.23 to 1.32) for the VacPulse group and an OR = 1.03 (95%CI 0.28 to 1.52) for the VacEPIG group, and they were not statistically different between vaccinated groups (Table 4).

## 4. Discussion

The results of this study demonstrate that the protection conferred by the administration of the trivalent vaccine with needle-free injection devices was not inferior compared to conventional needle-syringe. A common clinical feature of subclinical PCV2 infection and enzootic pneumonia is retardation of growth performance. Therefore, it is necessary to compare the growth performance for the evaluation of vaccination with needle-free injection devices and conventional needle-syringe injection. Trivalent vaccination of pigs with either needle-free injection devices or conventional needle-syringe injection improved growth performance compared to those of unvaccinated pigs. In addition, needle-free injection devices were not inferior to conventional needle-syringe injection for growth performance.

Approximately 30% and 20% of needle-free Pulse FX and needle-free EPIG injection device-vaccinated pigs had visible vaccine residue following vaccination, respectively. Residual vaccine that remains on the skin surface following vaccination with needle-free injection devices has previously been reported [20]. Adoption of needle-free vaccination may be delayed due to this unavoidable result. Comparable antibody responses against PCV2 and *M. hyopneumoniae* in pigs were measured in this study between two needle-free injection devices and the conventional needle-syringe injection. The needle-free injection device may allow greater penetration into the skin and greater dispersion in underlying tissue, resulting in this comparable immune response to conventional vaccination [21,22,23]. Data presented in this study suggests that frequent vaccine residue from needle-free injection device-vaccinated pigs may not influence the immune response. Nevertheless, administration with needle-free injection device requires a certain level of training of vaccination technique, and this may also influence the success of right administration and potentially reduce vaccine residue. Vaccine delivery with the small needle-free orifice and high pressure could potentially damage the vaccine’s antigenic component through nicking or degradation, thereby altering its antigenicity or immunogenicity. An equal immune response of trivalent vaccination between needle-syringe and needle-free devices indicated that this potential degradation of vaccine antigen did not occur.

For the immunological evaluation, trivalent vaccination of pigs with either needle-free injection devices or conventional needle-syringe injection induce significantly higher levels of PCV2 and *M. hyopneumoniae* antibodies compared to those of unvaccinated pigs. The immune response to trivalent vaccine given with the needle-free Pulse injection device was non-inferior to the immune response to trivalent vaccine given with conventional needle-syringe injection. For microbiological evaluation, the trivalent vaccination of pigs with needle-free devices was able to reduce PCV2 loads in blood and *M. hyopneumoniae* loads in larynx at equal levels with the conventional needle-syringe administration. The reduction in PCV2 viremia and *M. hyopneumoniae* laryngeal shedding is clinically significant information. Trivalent vaccination with either needle-free injection devices or conventional needle-syringe injection would decrease the risk of a potential horizontal transmission to other pigs by reducing the amount of PCV2 in circulation and *M. hyopneumoniae* shedding within the herd. During pathological evaluation, trivalent vaccination of pigs with needle-free injection devices was able to reduce severity of mycoplasmal-induced lung lesions and PCV2-associated lymphoid lesions at equal levels compared to conventional needle-syringe methods.

Needle-free injection device systems have many benefits for pigs and labor workers, as well as for consumers. In respect to pig welfare, it reduces the pig pain and stress caused by needle injection, leading to better general health and growth capabilities [6]. Swine workers experience enhanced safety and enhance productivity from needle-free injection device systems by eliminating the risk of accidental needle stick injuries and repetitive motion injuries attributed to manually squeezing syringe and improving the vaccination process [24]. Needle-stick injuries among swine veterinarians account for the highest number of physical injuries with 580 out of 794 surveyed veterinarians (73%) suffering needle-stick injuries [25]. In relation to food safety, needle-free injection device systems eliminate the possibility of broken needles and damage to the pig carcass [7]. This results in significant market benefits of the meat.

To the authors’ knowledge, this is the first field trial in a commercial swine operation to confirm the advantage of needle-free devices to administer vaccines intramuscularly over conventional needle-syringe vaccination technique while equating immune responses between the two. Vaccine administration via needle-free injector is less stressful for pigs, they stimulate an effective immune response, and are easy and safe for swine workers to use. The tested needle free injection devices were no inferior to needle-syringe in the effectiveness of pig vaccination offering all the benefits of needle-free vaccination.

## Figures and Tables

**Figure 1 vaccines-10-00358-f001:**
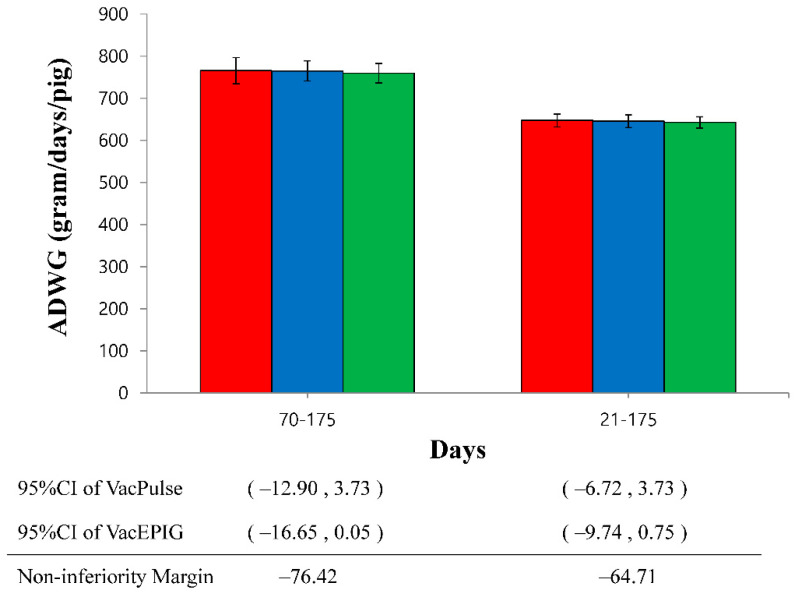
Non-inferiority analysis of average daily weight gain (ADWG) for 2 needle-free devices (VacPulse group ●, and VacEPIG group, ●) during fattening period (70 to 175 days) and overall period (21 to 175 days) as compared with conventional needle-syringe (VacS group, ●). The lower bound of the one-sided 95% confidence intervals (CI) was greater than the margin of non-inferiority, indicating that two needle-free devices were non-inferior to conventional needle-syringe.

**Figure 2 vaccines-10-00358-f002:**
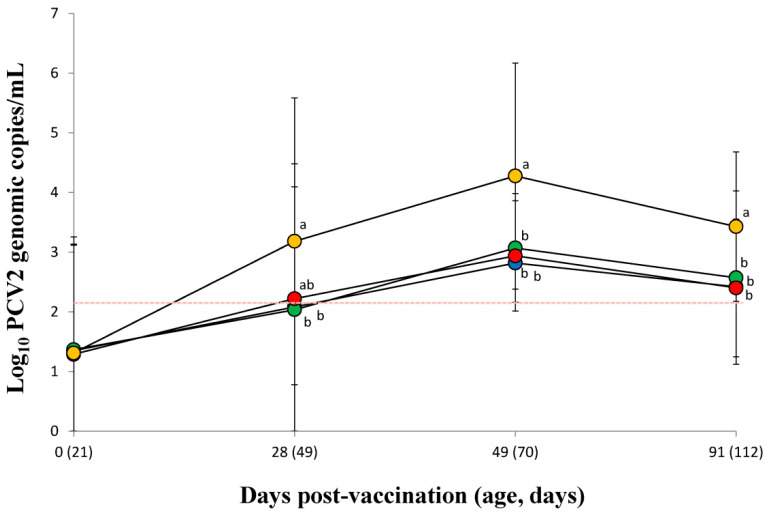
Mean values of the genomic copy number of PCV2d DNA in serum from VacS (●), VacPulse (●), VacEPIG (●), and UnVac (●) groups. Variation is expressed as the standard deviation. The detection limit of the assay is 1.3 × 10^2^ genomic copy numbers of PCV2d (red dotted line). Different superscripts (a and b) indicate significant (*p* < 0.05) different among 4 groups.

**Figure 3 vaccines-10-00358-f003:**
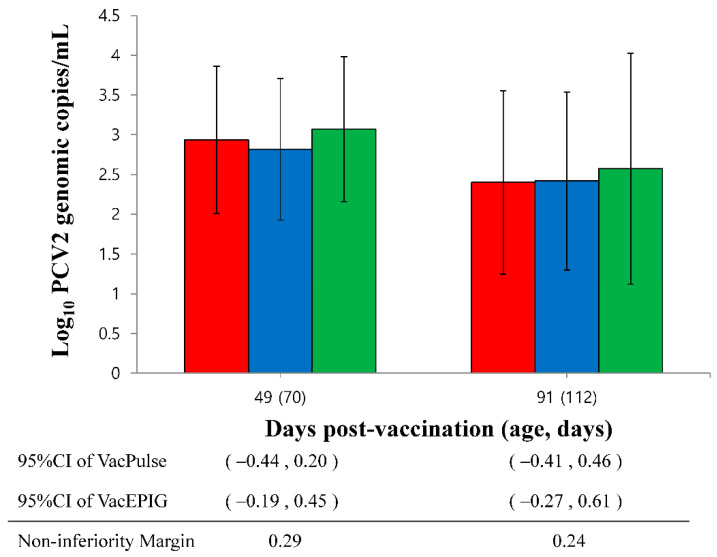
The upper bound of the 95% confidence intervals (CI) on the difference in PCV2d load in serum in the needle-free device (VacPulse, ●) group at 49 days post-vaccination (dpv) showed non-inferiority to conventional needle-syringe (VacS, ●) group. The PCV2d loads in serum of needle-free device (VacEPIC, ●) group at 49 and 91 dpv did not show non-inferiority to conventional needle-syringe (VacS, ●) group.

**Figure 4 vaccines-10-00358-f004:**
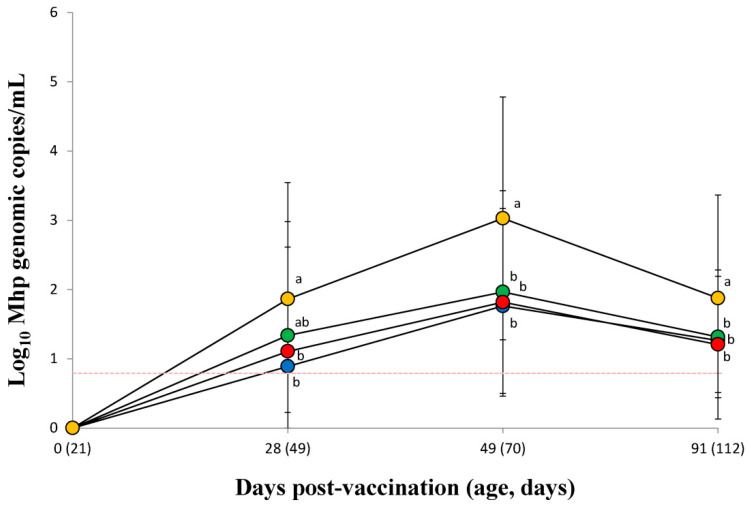
Mean values of the genomic copy number of *Mycoplasma hyopneumoniae* DNA in larynx from VacS (●), VacPulse (●), VacEPIG (●), and UnVac (●) groups. Variation is expressed as the standard deviation. The detection limit of the assay is 6.3 genomic copy numbers of *M. hyopneumoniae* (red dotted line). Different superscripts (a and b) indicate significant (*p* < 0.05) different among 4 groups.

**Figure 5 vaccines-10-00358-f005:**
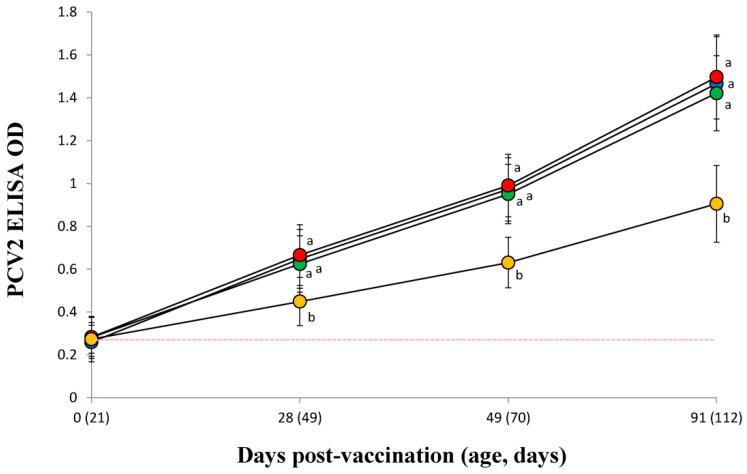
Mean values of the PCV2 antibodies by enzyme-linked immunosorbent assay (ELISA) in serum from VacS (●), VacPulse (●), VacEPIG (●), and UnVac (●) groups. Variation is expressed as the standard deviation. Serum samples are considered positive for PCV2 antibodies if the optical density (OD) is greater than 0.3 (red dotted line). Different superscripts (a and b) indicate significant (*p* < 0.05) different among 4 groups.

**Figure 6 vaccines-10-00358-f006:**
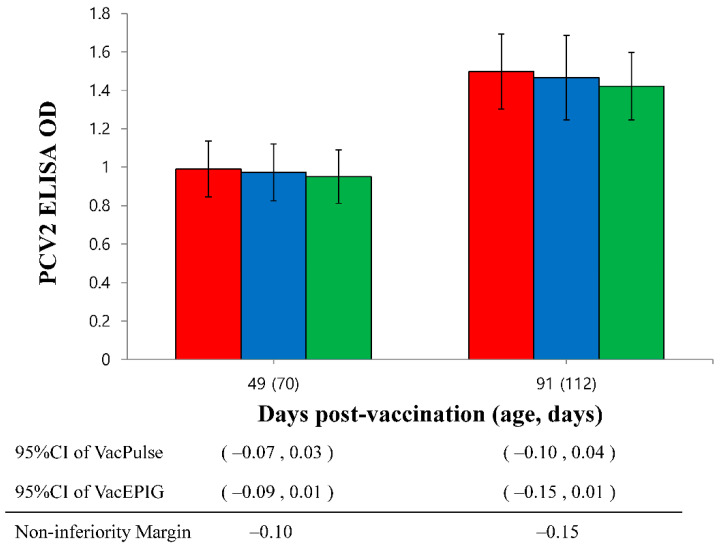
The lower bound of the 95% confidence intervals (CI) on the difference in porcine circovirus type 2 (PCV2) enzyme-linked immunosorbent assay (ELISA) in the needle-free device (VacPulse, ●) group at 49 days and 91 days post-vaccination (dpv) showed non-inferiority to conventional needle-syringe (VacS, ●) group. The lower bound of the 95%CI on the difference in PCV2 ELISA in the needle-free device (VacEPIG, ●) group at 49 dpv showed non-inferiority to conventional needle-syringe (VacS, ●) group.

**Figure 7 vaccines-10-00358-f007:**
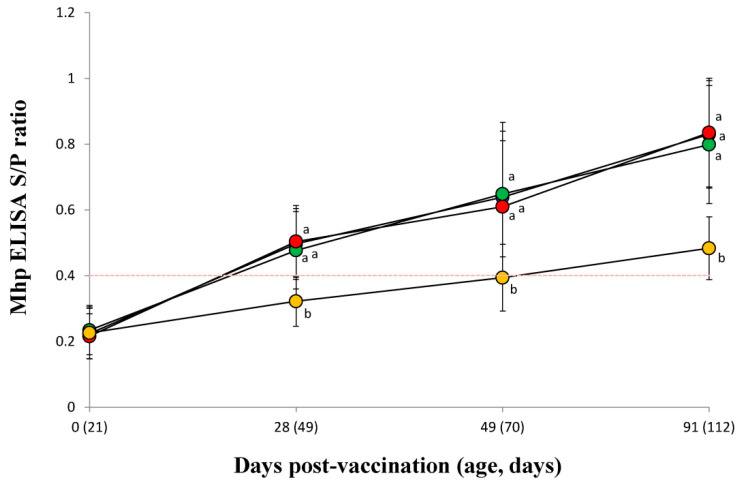
Mean values of the *Mycoplasma hyopneumoniae* antibodies by enzyme-linked immunosorbent assay (ELISA) in serum from VacS (●), VacPulse (●), VacEPIG (●), and UnVac (●) groups. Variation is expressed as the standard deviation. Serum samples are considered positive for *M. hyopneumoniae* antibodies if the sample-to-positive (S/P) ratio was ≥0.4 (red dotted line). Different superscripts (a and b) indicate significant (*p* < 0.05) different among 4 groups.

**Figure 8 vaccines-10-00358-f008:**
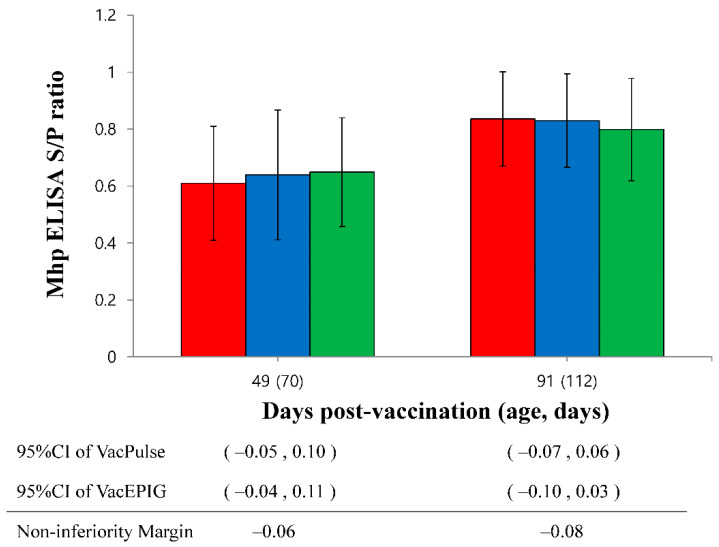
The lower bound of the 95% confidence intervals (CI) on the difference in *Mycoplasma hyopneumoniae* enzyme-linked immunosorbent assay (ELISA) in the needle-free device (VacPulse, ●) group at 49 days and 91 days post-vaccination (dpv) showed non-inferiority to conventional needle-syringe (VacS, ●) group. The lower bound of the 95%CI on the difference in *M. hyopneumoniae* ELISA in the needle-free device (VacEPIG, ●) group at 49 dpv showed non-inferiority to conventional needle-syringe (VacS, ●) group.

**Table 1 vaccines-10-00358-t001:** Field experimental design.

Groups	No. of Pigs	Injection Instrument	Dosage	Age (Days)
VacS	60	Syringe	One (2.0 mL)	21
VacPulse	60	Needle-free Pulse FX device	One (2.0 mL)	21
VacEPIG	60	Needle-free EPIG device	One (2.0 mL)	21
UnVac	60	Syringe	One (2.0 mL)	21

**Table 2 vaccines-10-00358-t002:** Body weight and average daily weight gain (ADWG) data (mean ± standard deviation) from vaccinated and unvaccinated groups.

	Age (Days)	Groups			
		VacS	VacPulse	VacEPIG	UnVac
Body weight	21	6.02 ± 0.33	5.96 ± 0.30	5.99 ± 0.32	5.93 ± 0.35
(Kg)	175	105.65 ± 2.46 ^a^	105.32 ± 2.24 ^a^	104.93 ± 1.99 ^a^	99.30 ± 2.68 ^b^
ADWG	21–70	392.82 ± 39.44	388.71 ± 31.96	391.87 ± 31.50	376.50 ± 33.16
(gram/pig/day)	70–175	765.21 ± 31.16 ^a^	764.78 ± 23.91 ^a^	759.41 ± 22.76 ^a^	713.04 ± 26.84 ^b^
	21–175	647.03 ± 15.32 ^a^	645.14 ± 15.04 ^a^	642.55 ± 13.12 ^a^	606.36 ± 17.86 ^b^

Different superscripts (a and b) indicate significant (*p* < 0.05) difference among 4 groups.

**Table 3 vaccines-10-00358-t003:** Macroscopic and microscopic pathology (mean ± standard deviation) of vaccinated and unvaccinated groups.

	Groups			
	VacS	VacPulse	VacEPIG	UnVac
Macroscopic	16.92 ± 9.72 ^a^	18.08 ± 10.34 ^a^	19.66 ± 9.87 ^a^	31.62 ± 11.05 ^b^
lung lesions				
Microscopic	0.73 ± 0.62 ^a^	0.82 ± 0.71 ^a^	0.92 ± 0.68 ^a^	2.06 ± 0.59 ^b^
lung lesions				
Microscopic	0.56 ± 0.57 ^a^	0.51 ± 0.47 ^a^	0.60 ± 0.52 ^a^	1.76 ± 0.63 ^b^
lymphoid lesions				

Different superscripts (a and b) indicate significant (*p* < 0.05) difference among 4 groups.

**Table 4 vaccines-10-00358-t004:** Odds ratio (OR) and 95% confidence intervals (CI) for pathological outcome.

	Macroscopic Lung Lesions		Microscopic Lung Lesions		Microscopic Lymphoid Lesions	
	OR	(95%CI)	OR	(95%CI)	OR	(95%CI)
VacPulse	0.96	(0.47–2.00)	1.34	(0.63–2.89)	0.56	(0.23–1.32)
VacEPIG	1.19	(0.57–2.46)	1.50	(0.70–3.24)	1.03	(0.28–1.52)
VacS	1.00	(Reference)	1.00	(Reference)	1.00	(Reference)

## Data Availability

The data present in the study are available on request from the corresponding author.

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
