# Peer review of "Non-Inferiority Field Study Comparing the Administrations by Conventional Needle-Syringe and Needle-Free Injectors of a Trivalent Vaccine Containing Porcine Circovirus Types 2a/2b and Mycoplasma hyopneumoniae"

_vaccines, 2022, doi:10.3390/vaccines10030358_

Round 1

Reviewer 1 Report

The clinical field trial was conducted on a 1,200-sow farrow-to-finish swine farm withan all-in-all-out production system. What breed of pig ?

The chosen farm status of porcine repro-ductive and respiratory syndrome (PRRS) was stable.

Please explain what means stable.

Author Response

Responses to Reviewer #1 comments

The clinical field trial was conducted on a 1,200-sow farrow-to-finish swine farm within all-in-all-out production system. What breed of pig ?
Response: Authors described the breed of pig in 2.1. Farm in Line 73.

The chosen farm status of porcine reproductive and respiratory syndrome (PRRS) was stable.

Please explain what means stable.

Response: Authors explained the meaning of stable of PRRS in 2.1. Farm in Line 76-77.

Reviewer 2 Report

In this study, the authors compared the administrations of trivalent vaccine with conventional needle syringe and needle-free syringe. The results showed that the protection conferred by the administration of the trivalent vaccine with needle-free injection devices was not inferior compared to conventional needle-syringe.

To date, vaccination with needle-free injection devices and conventional needle-syringe injection has been widely used in the field. However, this reviewer thinks that the injection method will not significantly affect the vaccination. Especially, the vaccination effect of needleless injection device has been proved in other vaccines, and there will be no great difference due to different vaccines. This manuscript lacks novelty.

Author Response

Responses to Reviewer #2 comments

In this study, the authors compared the administrations of trivalent vaccine with conventional needle syringe and needle-free syringe. The results showed that the protection conferred by the administration of the trivalent vaccine with needle-free injection devices was not inferior compared to conventional needle-syringe.

To date, vaccination with needle-free injection devices and conventional needle-syringe injection has been widely used in the field. However, this reviewer thinks that the injection method will not significantly affect the vaccination. Especially, the vaccination effect of needleless injection device has been proved in other vaccines, and there will be no great difference due to different vaccines. This manuscript lacks novelty.

Response: Authors agreed on the Reviewer #2 comments. However, it is important to conduct the field study to evaluate and compare the efficacy of conventional needle-syringe and needle-free syringe before farmers use. In addition, this study was conducted the comprehensive analysis of clinical, immunological, microbiological and pathological outcomes.

Reviewer 3 Report

In this manuscript the performance of the two needle-free methods were evaluated in the field, The results indicated that comparable ADVG, PCV2d viral copies and antibodies, Mycoplasma hyopneumoniae DNA and antibodies, and Macroscopic lung lesions were found among the immunization groups. The results were predictable, but still useful for the field disease control. However, there are some concerns that need to be clarified.

  1. As a farrow to finish swine farm, I wonder if the weaning place of the groups were comparable, for the temperature, wind and other factor will influence the ADVG and  weaning is the most important stage for pig industry.
  2. As discussed, the residual vaccine that remains on the skin was unavoidable, I wonder if  the 10% difference between group Plus FX and EPIG will influence the performance of the two methods, for there were some difference between the two groups in Figure 4 at 28DPV.
  3. Although it is understandable for the sera collection frequency, but 3 weeks is a defect and will influence the sensitivity of viral copies detection and antibody detection. There might be some difference of Mhp geuomic copies among the three groups between 28 and 49DPV.
  4. The cut line of the detection should be present.

Author Response

Responses to Reviewer #3 comments

In this manuscript the performance of the two needle-free methods were evaluated in the field, The results indicated that comparable ADVG, PCV2d viral copies and antibodies, Mycoplasma hyopneumoniae DNA and antibodies, and Macroscopic lung lesions were found among the immunization groups. The results were predictable, but still useful for the field disease control. However, there are some concerns that need to be clarified.

  1. As a farrow to finish swine farm, I wonder if the weaning place of the groups were comparable, for the temperature, wind and other factor will influence the ADWG and weaning is the most important stage for pig industry.

Response: Authors agreed on the reviewer #2’s comments. However, the all 4 groups have same field conditions so, there was no bias for the influence of ADWG among the 4 groups.

  1. As discussed, the residual vaccine that remains on the skin was unavoidable, I wonder if the 10% difference between group Plus FX and EPIG will influence the performance of the two methods, for there were some difference between the two groups in Figure 4 at 28DPV.

Response: It seemed that 10% differences between group Plus FX and EPIG will influence the performance of the two methods. However, statistically, there were no significant differences between group Plus FX and EPIG at 28 DPV. If the 10% difference between group Plus FX and EPIG influence the performance of the two methods, it should be significant differences between group Plus FX and EPIG at 49 DPV. In this study, there were no significant differences between group Plus FX and EPIG at 28 DPV.

  1. Although it is understandable for the sera collection frequency, but 3 weeks is a defect and will influence the sensitivity of viral copies detection and antibody detection. There might be some difference of Mhp genomic copies among the three groups between 28 and 49DPV.

Response: Since Mhp genomic copies were not detected among the three groups at 0 DPV, the results of Mhp genomic copies on 0 DPV do not influence the sensitivity of mycoplasma copies detection and antibody detection.

  1. The cut line of the detection should be present.

Response: Authors added the cut line of the detection in Figure 2 with line 242-243, Figure 4 with line 272-273, Figure 5 with line 287-288, and Figure 7 with line 310-311.

Round 2

Reviewer 2 Report

The pigs from the two needle-free injection device and conventional needle-syringe injection had significantly (p < 0.05) lower macroscopic and microscopic lung lesion scores, and microscopic lymphoid lesions than from unvaccinated. However, the vaccination of trivalent vaccine by the two needle-free Pulse injection devices used in the study was non-inferior to that by conventional needle-syringe injection for growth performance, immune response against PCV2 and M. hyopneumoniae, and reduction of PCV2 viremia. Why do you use the needle-free Pulse injection devices, but not conventional needle-syringe injection?

Numerous papers showed that needle free injection is better or equal to the needle-syringe injection, both in the lab or in the field. For example, Vaccine. 2021 Sep 15;39(39):5557-5562; Korean J Food Sci Anim Resour. 2018 Dec;38(6):1155-1159; Sci Rep. 2021 Nov 29;11(1):23107; Vaccine. 2021 Oct 29;39(45):6691-6699. Although this study conducted the comprehensive analysis of clinical, immunological, microbiological and pathological outcomes, it is still not inferior to the needle syringe injection in these aspects. I don't think that different vaccines with the same inoculation method (needle free injection) will affect the immune effect of the vaccine. Therefore, this manuscript lacks novelty.

Specific comments:

  1. The paragraph, line 155-165, is confused. Rewrite it.
  2. Line 184-191, substantial amounts of visible residue remained on the skin at the site of vaccination of both needle-free injection, but no visible vaccine residues in VacS group. Why? Will this increase the cost for the needle free group? Moreover, the skin reactions were worse in the needle free group. WHY?
  3. combine section 3.6 and 3.7
  4. Line 362-364, needle free injection is better for vaccine in penetration, dispersion and immune stimulation? Add more reference to prove it.
  5. what about the cost of the needle free injection and conventional vaccination? Which one is cheap and convenient for the farmer?

Author Response

Specific comments:

  1. The paragraph, line 155-165, is confused. Rewrite it.

Response: Authors rewrote the paragraph in Line 155-166.

  1. Line 184-191, substantial amounts of visible residue remained on the skin at the site of vaccination of both needle-free injection, but no visible vaccine residues in VacS group. Why? Will this increase the cost for the needle free group? Moreover, the skin reactions were worse in the needle free group. WHY?

Response: The skin reactions were not worse in the needle free group. The only disadvantage of needle free injection is to observed visible residues. However, the small amounts of visible residues on the skin at the site if vaccination are unavoidable due to injection system. Nevertheless, the efficacy of vaccine between needle free injection and syringe injection is not significantly different. Authors described the skin reaction in Lines 183-191.

  1. combine section 3.6 and 3.7

Response: section 3.6 is “Quantification of M. hyopneumoniae DNA in Larynx” and section 3.7 is “PCV2 serology”. Therefore, it is very difficult to combine section 3.6 and 3.7 because these two sections are different topics.

  1. Line 362-364, needle free injection is better for vaccine in penetration, dispersion and immune stimulation? Add more reference to prove it.

Response: Authors added more reference in Line 362-365.

  1. what about the cost of the needle free injection and conventional vaccination? Which one is cheap and convenient for the farmer?

Response: Frankly speaking, authors do not know the cost of the needle free injection because the vaccine company provide needle free injector if the farmers use the commercial vaccine from the same vaccine company. Also, it is very difficult to estimate the cost of conventional vaccination because the labor cost is different from different farms. Authors described the benefit for the needle free injection in Lines 390-400. Based on the discussion in Lines 390-400, the needle free injection is convenient for the farmer because the labor cost is getting increased and the lack of labor workers to be employed.

Round 3

Reviewer 2 Report

the manuscript has been sufficiently improved to publish in Vaccines.